# Differences in the Chloroplast Genome and Its Regulatory Network among *Cathaya argyrophylla* Populations from Different Locations in China

**DOI:** 10.3390/genes13111963

**Published:** 2022-10-27

**Authors:** Kerui Huang, Ping Mo, Aihua Deng, Peng Xie, Yun Wang

**Affiliations:** Hunan Provincial Key Laboratory for Molecular Immunity Technology of Aquatic Animal Diseases, College of Life and Environmental Sciences, Hunan University of Arts and Science, Changde 415000, China

**Keywords:** *Cathaya argyrophylla*, chloroplast genome, endangerment, regulatory network, physiological sensitivity

## Abstract

*Cathaya argyrophylla* Chun et Kuang is a severely endangered, tertiary relict plant unique to China whose high physiological sensitivity to the environment, including photosensitivity, is likely closely related to its endangered status; however, the exact mechanism responsible has remained unknown due to the rarity of the plant and the difficulties involved in performing physiological studies on the molecular level. In this study, the chloroplast genomes of six *C. argyrophylla* populations sampled from different locations in China were characterized and compared. In addition, a gene regulatory network of the polymorphic chloroplast genes responsible for regulating genes found elsewhere in the plant genome was constructed. The result of the genome characterization and comparison showed that the genome characteristics, the gene composition, and the gene sequence of the chloroplast genes varied by location, and the gene regulatory network showed that the differences in growth location may have led to variations in the protein-coding chloroplast gene via various metabolic processes. These findings provide new insights into the relationship between chloroplasts and the sensitive metabolism of *C. argyrophylla* and provide additional reference materials for the conservation of this endangered plant.

## 1. Introduction

*Cathaya argyrophylla* Chun et Kuang is a tertiary relict plant unique to China that has important research and practical value [1,2]. It is partially distributed in the southern Hunan Province, including the Bamian Mountain near Zixing City and the Yuechengling Mountains of Chengbu County. Other populations are also located in the Guangxi, Guizhou and Chongqing provinces. *C. argyrophylla* is listed as a plant species with extremely small populations in China and is severely endangered due to the extremely high mortality rate of *C. argyrophylla* plants and seedlings in the wild [3], dramatic reductions in its distribution and yearly decreases in the number of wild populations. It is thought that the endangered status of this species is closely related to its extreme physiological sensitivity. Since *C. argyrophylla* originated in high-altitude mountains on steep peaks and ridges, it is very sensitive to the environment and tends to grow weakly or not survive in environments that are highly differentiated from its natural habitat, which has resulted in a very low introduction success rate [4]. A better understanding of the mechanisms responsible for *C. argyrophylla*’s extreme environmental sensitivity is urgently needed, given the critical status of this species [3].

It has been long reported that *C. argyrophylla* is sensitive to light and requires certain shade conditions during its seedling stage, with too much shade or intense light leading to plant death [5]. Some studies have recently reported that the photosensitivity of *C. argyrophylla* plays an important role in its distribution, with its high light compensation and light saturation points causing its narrow distribution area [6]. Other physiological and ecological studies have shown that its low photosynthetic capacity is related to its critically endangered status [7,8], but the underlying mechanism responsible for its photosensitivity and environmental sensitivity remains unknown, which requires urgent research.

The chloroplast is a unique plastid that is of great significance to the energy metabolism of plants [9]. Chloroplasts have their own genome and a very conservative structure, content, gene sequence and base composition. Chloroplasts contain about 130 genes, including about 30 transfer RNAs, 30 genes related to gene expression and 50 genes related to photosynthesis [10]. Most photosynthesis-related genes encoded by the chloroplast are crucial to the metabolic processes that occur within the chloroplast and the entire plant. Abnormal gene expression leads to abnormal chloroplast development and even plant death. For example, when the *petA* gene is knocked out, the abnormal function of the cytochrome complex-related genes *petN* and *petG* lead to albino traits and growth retardation [11,12]. It has also been found that the abnormal expression of the enoyl-acyl carrier protein reductase in chloroplasts can lead to the accumulation of reactive oxygen species, which can further induce programmed cell death [13]. Given the importance of the chloroplast in photosynthesis and plant metabolism, a better understanding of the role that chloroplasts play in the relationship between the endangered status of *C. argyrophylla* and its photosensitivity is warranted, and urgent research is needed.

In this study, the complete chloroplast genomes of five *C. argyrophylla* populations from China were sequenced and characterized, then compared with each other and an additional population from the Taiwan Province from a previous study [14]. In addition, a gene regulatory network based on machine learning algorithm was constructed to predict the relationship between the specimens’ polymorphic chloroplast genes and the overall plant metabolism. Our findings provide new insights into the relationship between chloroplasts and the physiological metabolism of *C. argyrophylla* and provide additional reference materials for the conservation of this endangered plant.

## 2. Materials and Methods

### 2.1. Plant Materials

Specimens were collected from five naturally grown *C. argyrophylla* populations found in the Yuechengling Mountains of Chengbu County (26°33′12.14″ N 110°34′52.30″ E at 619 m for the sample CBRY2; 26°32′31.48″ N 110°34′49.45″ E at 928 m for the sample CBTY1; 26°32′18.37″ N 110°34′47.75″ E at 1066 m for the sample CBTY2) and on the Bamian Mountain in Chengzhou City (26°4′8.88″ N 113°42′59.51″ E at 1137 m for the sample SMP; 26°3′23.59′N 113°42′55.92″E at 1213 m for the sample JPL) in the Hunan Province (Figure 1). Five fresh leaves were collected from a mature plant for each location, then dried for DNA extraction. The voucher specimens were preserved at the College of Life and Environmental Sciences, Hunan University of Arts and Sciences (contact person: Kerui Huang, huangkerui008@163.com).

### 2.2. Chloroplast Genome Sequencing and Annotation

Total genomic DNA was extracted from the frozen leaves using a Dneasy plant tissue kit (TIANGEN Biotech Co., Ltd., Beijing, China). The library was constructed using total DNA and sequenced using the Illumina HiSeq 2500 platform (Shanghai Personalbio Technology Co., Ltd., Shanghai, China). The reads were retained after filtering out low-quality reads with fastp v0.23.1 [15]. Subsequently, de novo assembly of the *C. argyrophylla* chloroplast genome was performed using GetOrganelle v1.7.5 [16] (accessed on 12 January 2021) and 5,000,000 reads were randomly selected to reduce the computational time and resources. The Plastid Genome Annotator [17] was used to annotate the chloroplast genome.

### 2.3. Sequence Variation Analysis

Six *C. argyrophylla* chloroplast genomes were selected to compare the genomic structure, including five individuals from the Hunan Province collected during this study and one individual from Taipei in the City of Taiwan Province (hereinafter abbreviated as TW), previously published by the National Center for Biotechnology Information (NCBI) with the accession number of NC 014589. All genomes were compared using the mVISTA software [18] with the genome from CBTY2 as a reference. Chloroplast genes for each genome were extracted using Geneious Prime v2022.0.1 and were aligned using MAFFT v7.313 to obtain the polymorphic chloroplast genes from each sample.

### 2.4. Phylogenetic Analysis

The phylogenetic analysis was performed using the common protein-coding genes of each sample. First, each gene was aligned individually using MAFFT v7.313, and those with polymorphic sites were selected for the tree construction. These genes were then linked end-to-end to form supergenes. Bayesian Inference phylogenies were constructed using MrBayes 3.2.6 (http://nbisweden.github.io/MrBayes/, accessed on 11 October 2021) under HKY+F model (2 parallel runs, 2,000,000 generations), in which the initial 25% of sampled data were discarded as burn-in.

### 2.5. Gene Regulatory Network Construction

Since *Arabidopsis thaliana* is a widely used model plant with abundant gene expression data available in public databases, we used the gene expression data of *A. thaliana* to construct a gene regulatory network of *C. argyrophylla* chloroplast genes that interact with genes found elsewhere in the *C. argyrophylla* genome. To obtain all the homologous chloroplast genes of *C. argyrophylla* and their literature-reported related chloroplast genes, the chloroplast genes obtained during the sequence annotation analysis were put into the STRING v11.0 database (accessed on 12 April 2022) to search for related genes in *A. thaliana*.

We used an expression-based machine learning network inference algorithm called MERLIN (modular regulatory network learning with per gene information) [19] to infer the regulatory network. Briefly, the *A. thaliana* expression data was downloaded from the Expression Atlas (https://www.ebi.ac.uk/gxa/experiments/E-GEOD-64740/Downloads, accessed on 11 October 2021) and entered into MERLIN. The data matrix had 268 samples from nine projects. From these, the mean for each gene was calculated, and the expression levels of the genes were zero-mean transformed. We kept only the genes whose expression value deviated by at least ±1 from the mean in at least five samples. The final input included 121 factors (chloroplast-related homologous genes of *C. argyrophylla*) and 21,638 targets. To assess the confidence of our inferred network connections, we created six sub-sets of the final data matrix, which each randomly contained 50% of the samples from the complete matrix. Each sub-set data was selected to infer a MERLIN network. The edges were selected for the final network on the condition that the edge appeared at least five times (confidence of 83.3%) in the six results. The final network connected 94 regulators to 4397 target genes. These results were exported into Cytoscape v3.8.1 to draw the final network.

Functional enrichment analyses (GO and KEGG) were conducted for the target genes using the STRING v11.0 database. A two-tailed Fisher’s exact test was used to test the enrichment of the target genes against all genome genes. Correction for multiple hypothesis testing was carried out using standard false discovery rate control methods. The GO or KEGG terms with a corrected *p*-value of < 0.05 was considered significant.

## 3. Results

### 3.1. Comparison of Chloroplast Genomic Features of C. argyrophylla

The features of the obtained sequences and their accession numbers in the NCBI database are listed in Table 1. Figure 1 provides the location of each *C. argyrophylla* population. The physical map was drawn from the annotation GB file of CBTY2 (Figure 2).

Table 1 shows the *C.*
*argyrophylla* chloroplast genome sizes grown in the six locations: 118,538 bp (CBTY1); 118,886 bp (CBTY2); 120,328 bp (CBRY2); 118,724 bp (JPL); 119,080 bp (SMP); and 107,122 bp (TW). The shortest chloroplast genome was from TW. The size of all IR regions was relatively short, with only around 850 bp for all samples, with TW being the shortest at only 782 bp. All samples had a GC content of 39%, similar to the LSC and SSC content at around 38% and 39%, respectively. The GC content for the IR region ranged from 34% (CBTY1) to 39% (CBTY2 and SMP). These values were lower than those of LSC and SSC, which indicates the low number of rRNA and tRNA genes in the IR regions [20,21]. Each genome had a similar number of total genes, CDSs and rRNA genes. The TW genome contained the least amount of tRNAs, which may be responsible for its smaller genome size. Thus, the overall features of the TW chloroplast genome were quite different from those of the five populations from China.

Although the number of chloroplast genes in each sample was similar (Table 1), their gene compositions differed greatly (Figure 2 and Figure 3), especially when TW was compared to the remaining locations. There were 112, 108, 112, 109, 109 and 110 genes for CBTY1, CBTY2, CBRY2, JPL, SMP and TW, respectively. There were 22 genes unique to TW, of which 21 were tRNA genes (such as *TrnC-TrnI*, *TrnK* and *TrnL*) and one was a protein-coding gene (clpP). There were 32 genes unique to the five remaining populations located in the Chinese hinterlands, of which 31 were tRNA genes (such as *TrnC-GCA*, *TrnD-GUC* & *TrnE-UUC*) and one was a protein-coding gene (*rps7*). Thus, the TW chloroplast gene composition was quite different from the remaining samples (CBTY1, CBTY2, CBRY2, JPL & SMP; Figure 3).

There were also some slight differences between the hinterland samples. For example, the CBTY2, JPL and SMP samples were clustered together, while CBRY2 and CBTY1 were clustered together (Figure 3). This differentiation was caused mainly by the *psbM* gene, which was found in CBRY2 and CBTY1, but was absent in CBTY2, JPL and SMP.

### 3.2. Comparison of C. argyrophylla Chloroplast Genome Sequence Variations

The genomic sequence variations were compared between the different locations (Figure 4), given the location-based differences in the chloroplast genomes. CBTY2 was used as a reference sequence since its features were similar to JPY and SMP. The results were quite consistent with the gene composition comparison in that the CBTY2, JPL and SMP genome sequences were visibly similar to those of the other samples. Similarly, CBTY1 and CBRY2 were found to have much in common, with the TW genome sequence being quite different from all other samples. This was especially the case for the SSC genes, which had several genes (such as *ycf2*) that were missing fragments of considerable length. This might have been caused by the different sequencing and assembly methods used between this study and that of Lin et al. [14].

Homologous protein-coding chloroplast genes from the *C. argyrophylla* samples were extracted, and then sequence alignments were performed to further clarify the sequence variations among the various locations. As a result, 18 chloroplast protein-coding genes (except *ycf2*, which may be due to a technical error in another study) with polymorphic sites were selected (Figure 5). These included a variety of genes, such as genes for the photosystem, cytochrome b6/f complex, ATP synthase, Rubisco, RNA polymerase, ribosomal proteins and maturase (Table 2). According to their polymorphic sites and mutations types, these genes could be divided into four groups: (1) those with identical and unique polymorphic sites and mutation types from the SMP and JPL samples (CZ group); (2) those with unique polymorphic sites and mutation types from TW and those with no polymorphic sites from the remaining hinterland groups (TW group); (3) those with unique polymorphic sites and mutation types from TW and those with polymorphic sites from the remaining hinterland groups (TW2 group) and (4) those with various polymorphic sites that were difficult to organize (ELSE group). There were 2, 10, 2 and 4 genes for the CZ, TW, TW2 and ELSE groups, respectively. TW contained 66.7% of all the polymorphic genes, which indicates that the sequence polymorphisms of most chloroplast genes in this study mainly resulted from the uniqueness of the TW sample.

### 3.3. Phylogenetic Analysis of the C. argyrophylla Populations in Different Locations

Phylogenetic analysis of the *C. argyrophylla* populations was carried out using the end-to-end linkage of the polymorphic chloroplast genes among the *C. argyrophylla* chloroplast genomes studied above. The result (Figure 6) showed that the genetic distance between the *C. argyrophylla* population from Taiwan city and those from the hinterland is relatively huge compared to those between the hinterland populations, which is consistent with the results above and makes the Taiwan population unique to other populations. However, the phylogenetic tree did not separate the populations well by locations in the hinterland of China. Thus, it seemed to be difficult to distinguish *C. argyrophylla* populations by location using chloroplast genome-based phylogenetic methods.

### 3.4. Gene Regulatory Network Analysis of the Polymorphic Chloroplast Genes

Chloroplast genes are essential for the survival of many plants, especially for the endangered *C. argyrophylla* with its low photosynthetic capacity and photosensitivity [5,6,7,8,9]. Given the high possibility that the locations of *C. argyrophylla* populations affect their chloroplast gene compositions and sequences, it was hypothesized that the 18 location-related polymorphic chloroplast genes either mediated or were signs of the physical differences found between these populations. To clarify this hypothesis and better understand the mechanism responsible, a regulatory network of all chloroplast genes that regulated genes found in the rest of the plant genome was constructed using the machine learning algorithm MERLIN (Roy et al., 2013) based on the gene expression atlas of 268 *A. thaliana* samples. The final constructed network is shown in Figure 7. The network contained regulations between 93 regulators (i.e., chloroplast genes, which mainly existed along the inner circle of the network, including the location-related polymorphic chloroplast genes and their related chloroplast genes as reported by the literature) and 4397 targets (i.e., regulated genes, which mainly existed along the outer ring of the network). This finding indicates that the chloroplast genes of *A. thaliana* are quite related to genes found in the remaining plant genome. The network showed that a considerable proportion of the interactions are regulated by those related to polymorphic, homologous chloroplast genes of *C. argyrophylla* (40.5%), among which most interactions were regulated by genes related to the CZ group. There was almost no intersection of the regulations guided by genes related to the CZ, TW, TW2 and ELSE groups, indicating great differences in the regulatory programs of the different *C. argyrophylla* polymorphic chloroplast homologous gene groups. The network’s Venn diagram (Figure 8) showed that over 50% of all chloroplast genes (50) were found in the inner circle of the network and that nearly half (5) of the hub genes of the network were related to these polymorphic, homologous chloroplast genes. This finding indicates the important role that most of the chloroplast genes and the polymorphic, homologous chloroplast genes play in the metabolism processes of *A. thaliana*, which may be similar for *C. argyrophylla*.

Functional annotation and enrichment analysis were conducted to explore the metabolic processes related to or mediated by the polymorphic, homologous chloroplast genes of each *C. argyrophylla* group (CZ, TW, TW2 and ELSE) in *A. thaliana*. Figure 7 shows the enrichment results as a circular barplot around the outermost portion of the network and shows that the regulated targets had large functional differences related to the different groups. For example, the CZ group’s target genes were regulated by *ptf1*, which was related to the *psbD* gene of the CZ group, and significantly responded to chemical stimulation, such as alcohol, abscisic acid and acid chemical. In comparison, the target genes regulated by the *pde247* gene, which was related to the *rpoA* gene of the CZ group, were significantly enriched in terms of gene silencing and plant development, such as ‘gene silencing by RNA’ and ‘shoot system development’. The target genes regulated by the *hcf163* and *sigE* genes, which were related to the *psbD* gene of the CZ group, were significantly enriched in terms of carotenoid biosynthesis and anion binding, respectively. The target genes of the TW group regulated by the *rhon1* gene, which was related to the *rbcL* gene of the CZ group, were significantly enriched in terms of gene expression, protein folding and plastid organization. The target genes of the TW2 group regulated by the *At5g42310* gene, which was related to the *petD* and *petB* genes of the TW2 group, were significantly enriched by molecular metabolism and light stimulation responses, such as ‘small molecule biosynthetic process’ and ‘cellular response to light stimulus’. Target genes regulated by the *alb3* gene, which were related to the *psbI* gene of the TW2 group, were significantly enriched by thylakoid. The large functional differences among the target genes regulated by the different group-related regulators (CZ, TW, TW2 and ELSE) indicate that the grouping of the *C. argyrophylla* polymorphic chloroplast genes was correct and further support the hypothesis that these 18 polymorphic chloroplast genes might mediate or signal the physical differences found among the studied *C. argyrophylla* populations. The notable metabolism processes might be related to light stimulation, chemical stimulation, plant development, gene silencing, carotenoid biosynthesis, gene expression, protein folding and plastid organization, molecular metabolism, and light stimulation responses.

The chloroplast gene interactions were extracted from the entire network to clarify the regulatory relationships between the *A. thaliana* chloroplast genes (Figure 9). No direct relationship was found between the genes of the different groups (CZ, TW and TW2). Furthermore, it was rare for a gene to be simultaneously regulated by genes related to multiple groups. This result indicates that the chloroplast genes in the different groups have quite independent regulatory mechanisms, which is consistent with the findings for the entire network. We also found that the regulatory ability of the location-related, polymorphic, homologous *C. argyrophylla* chloroplast genes was extremely low when judged by the number of targets (the degree), yet their literature-reported related chloroplast genes (genes in the lower case of Figure 9) have a high regulatory ability. Further, in the network obtained in this study, none of the location-related polymorphic chloroplast genes interacted with their literature-reported chloroplast genes. Thus, it is more likely that these location-related polymorphic chloroplast genes resulted from the different location-guided metabolisms rather than regulating them.

## 4. Discussion

The endangered status of *C. argyrophylla* is closely related to its physiological sensitivity to the environment (e.g., photosensitivity) [5,6,7,8]; however, studying the species on the molecular, physiological level has been difficult due to the plant’s rarity. Thus, the physiological basis of this species’ endangered status remains unclear, as does the related role of the chloroplast. In this study, we characterized and compared the *C. argyrophylla* chloroplast genomes between different *C. argyrophylla* populations in China and constructed a gene regulatory network of chloroplast genes to elucidate the relationship between the physiological sensitivity of *C. argyrophylla* and its chloroplasts.

### 4.1. Location Differences Lead to Differences in Chloroplast Genomes in C. argyrophylla

The findings of this study are consistent with those of previous studies that have reported differences in the chloroplast genomes within a single species, especially among species with populations found at differing locations [22,23,24]. Specifically, we found significant differences in the characteristics and sequences of the chloroplast genomes of six different *C. argyrophylla* populations (Figure 2, Figure 3, Figure 4 and Figure 5). A total of 18 chloroplast genes with polymorphic sites were selected and divided into four groups (CZ, TW, TW2, ELSE). The TW group had the greatest sequence variation (66.7%) compared to the remaining groups. This finding indicates that the sequence polymorphisms of the majority of the polymorphic chloroplast genes in this study primarily resulted from the uniqueness of the TW sample. Taiwan is a province and an island located off the southeast China (Figure 1). Most of the island has a tropical climate warmer than most hinterland areas of China throughout the year [25], which is remarkably different from the climate found in the other groups [25,26]. Thus, the great variation in this study’s chloroplast genomes of the TW group indicates that the Taiwanese climate may affect chloroplast gene composition and greatly influence the chloroplast gene sequences in *C. argyrophylla* plants. The results of the remaining groups also suggest that location may affect the chloroplast gene compositions and sequences in *C. argyrophylla*. For example, the gene map found in this study (Figure 2) clustered JPL & SMP together. Since JPL & SMP were both collected from Zixing County of Chenzhou City in the eastern Hunan Province, this result indicates that gene composition may also be related to location among the hinterland samples.

In a phylogeographic study by Wang and Ge (2006) [27], it was suggested that there are at least four separate glacial refuges for *C. argyrophylla* based on mitochondrial DNA. Wang and Ge (2006) [27] also found almost no long-distance dispersal or population expansion of this species at each location, which included Chenzhou City and Zixing County in the current study. We found that certain similarities between the chloroplast genome characteristics, sequences and polymorphisms of a few genes located in the SMP and JPL samples, which were both located in Chenzhou County, were unique from the samples from the other locations. These results support the conclusion of Wang and Ge (2006) [27] that the *C. argyrophylla* population located in Chenzhou City has unique gene sequence patterns; however, the sequence and the polymorphisms between the chloroplast genes found in the remaining locations, especially Chengbu City (CBTY1, CBTY2 & CBRY2) varied greatly, and also, the phylogenetic analysis in this study is not consistent with the study of Wang and Ge, which indicates that the chloroplast genome of *C. argyrophylla* may be more easily influenced by environmental conditions compared to the mitochondrial genome, which requires further study.

### 4.2. Gene Regulatory Network Revealed Potential Physiological Differences among C. argyrophylla Populations

MERLIN is a machine learning algorithm that infers regulatory networks between genes based on gene expression data and Bayesian networks using a probabilistic graphical model for network construction [19]. MERLIN has extremely high accuracy and is superior to many other similar algorithms. Only a few biological studies have used this algorithm. For example, Marx et al. used the MERLIN algorithm to study the relationship between *Medicago truncatula* and *Sinorhizobium meliloti* and found that calmodulin is a key regulator in the host species, *M. truncatula* [28]. In this study, MERLIN was used to predict the regulatory mechanisms of homologous *C. argyrophylla* polymorphic chloroplast genes and other genome-wide genes based on the expression data of *A. thaliana*.

The MERLIN-constructed network in this study contained 93 regulators and 4397 targets (Figure 7), with more than 50% of all chloroplast genes (50) displaying a high regulatory ability. This result indicates that the *A. thaliana* chloroplast genes were closely associated with other genome-wide genes, indicating a strong relationship between the chloroplast genes and the whole plant metabolism process for most plants. Although it has been previously proven that matter and energy are exchanged between the chloroplast and the remaining parts of the cell [29,30], the relationship between the chloroplast genes and the genes located in other parts of the cell was uncovered and confirmed by this study.

Moreover, a considerable proportion of the interactions found in the network (Figure 7) were regulated by genes associated with the polymorphic chloroplast genes of *C. argyrophylla* (40.5%) and nearly half (five) of the hub genes were related to these polymorphic chloroplast genes. This result indicates the essential relationship between the *C. argyrophylla* polymorphic chloroplast genes (and their related genes) and the metabolism of this species. In addition to the number of interactions between the polymorphic chloroplast genes and the genes found elsewhere in the plant’s genome, there was almost no intersection among the regulatory genes between the four groups (CZ, TW, TW2 and ELSE). This finding indicates a significant difference in the regulatory mechanisms between the different *C. argyrophylla* polymorphic chloroplast genes.

Functional enrichment analysis showed that the targets regulated by the genes from the different groups had large functional differences. For example, target genes regulated by chloroplast genes in the CZ group were mainly enriched in functional terms by chemical stimulation, plant development, gene silencing, carotenoid biosynthesis and anion binding. Since plant response to chemical stimulation often occurs when plants adapt to adverse soil conditions and carotenoid biosynthesis is also related to soil conditions [31,32,33], we hypothesize that the unique polymorphic *C. argyrophylla* chloroplast genes in Chengzhou City resulted from the effect of that location’s unique soil conditions. In addition, we found that genes regulated by the TW group were mainly enriched in functional terms by gene expression, protein folding and plastid organization. The genes regulated by the TW2 group genes were enriched primarily by molecular metabolism, light stimulation response and thylakoid. Since protein folding is a crucial method by which plants regulate their biological activity and is easily affected by temperature, especially heat stress [34,35,36] and light is an important abiotic condition for plant photosynthesis, which is also often related to temperature [37,38], we hypothesize that the target genes regulated by the TW and TW2 groups are the result of the warmer climate in the Taiwan Province. Since group classification was according to location and sequence variation, we hypothesize that the great variation and functional differences found in the genes regulated by genes in the different groups were caused by the differences in location. This hypothesis is further supported by the finding that the target genes were not directly regulated by the polymorphic chloroplast genes, but rather by their literature-reported related genes (Figure 9). Combining these results, we hypothesize that the unique polymorphisms of the chloroplast genes of the *C. argyrophylla* populations in the Taipei City of Taiwan Province were caused by Taiwan’s warmer climate, and the unique polymorphisms for those in Chengzhou City resulted from Chenzhou’s unique soil conditions. It is quite possible that certain physiological differences among *C. argyrophylla* populations located in the Hunan Province and the Taiwan Province are caused by their soil and climate conditions. This is an area in need of further study.

## 5. Conclusions

The physiological sensitivity of *C. argyrophylla* to the environment has remained a mystery given the plant’s rarity and the resulting difficulty in performing molecular physiological studies on this species. In this study, we characterized and compared the chloroplast genomes of *C. argyrophylla* from six different populations in China and constructed a gene regulatory network. We confirmed that the chloroplast genome characteristics and gene composition of *C. argyrophylla* are sensitive to plant location and climate. We also found that the differences in environment between different locations can lead to variations in the chloroplast protein-coding gene through variations in metabolic processes. Our findings provide new insights into the relationship between chloroplasts and the physiological metabolism of *C. argyrophylla* and provide additional reference materials for the conservation of this endangered plant.

## Figures and Tables

**Figure 1 genes-13-01963-f001:**
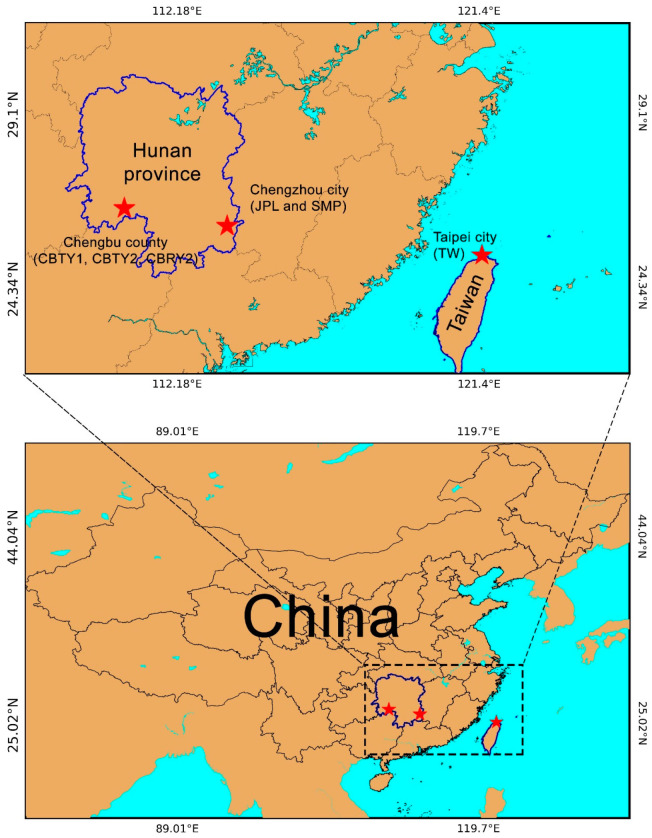
The locations of *Cathaya argyrophylla* populations sampled for this study. Stars indicate different cities or counties. Sample IDs are provided in parentheses.

**Figure 2 genes-13-01963-f002:**
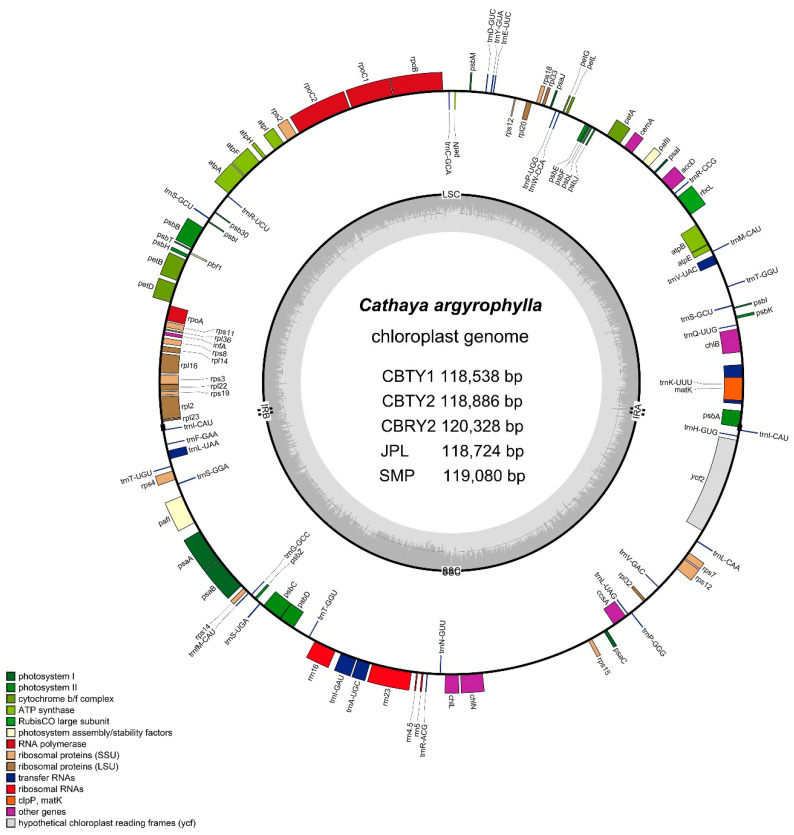
Gene map of the *Cathaya argyrophylla* chloroplast genome. The genes located within the circle were transcribed clockwise, while those outside were transcribed counterclockwise. The colors correspond to genes with different functions: the darker gray in the inner circle corresponds to DNA G + C content, while the lighter gray corresponds to A + T content. CBTY1, CBTY2 and CBRY2 samples were collected from Chengbu County of Hunan Province. JPL and SMP samples were collected from Chengzhou City of Hunan Province.

**Figure 3 genes-13-01963-f003:**
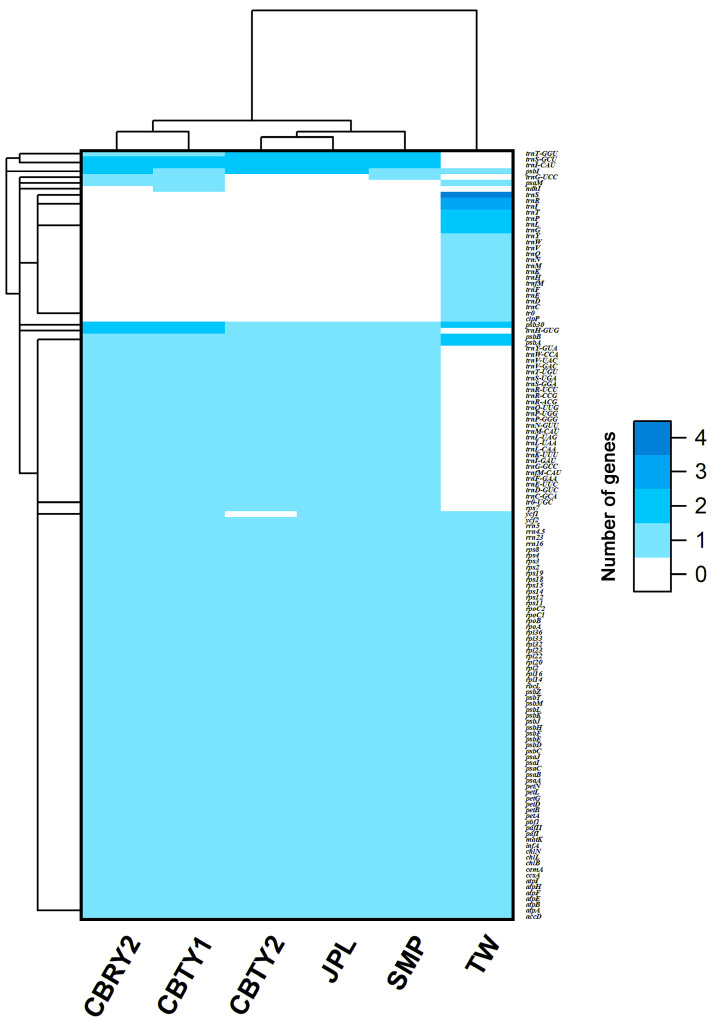
Comparison of chloroplast gene composition between *Cathaya argyrophylla* populations from different locations in China. CBTY1, CBTY2 and CBRY2 were collected from Chengbu County of Hunan Province. JPL and SMP were collected from Chengzhou City of Hunan Province. TW was collected from Taipei City of Taiwan Province in a previously published study.

**Figure 4 genes-13-01963-f004:**
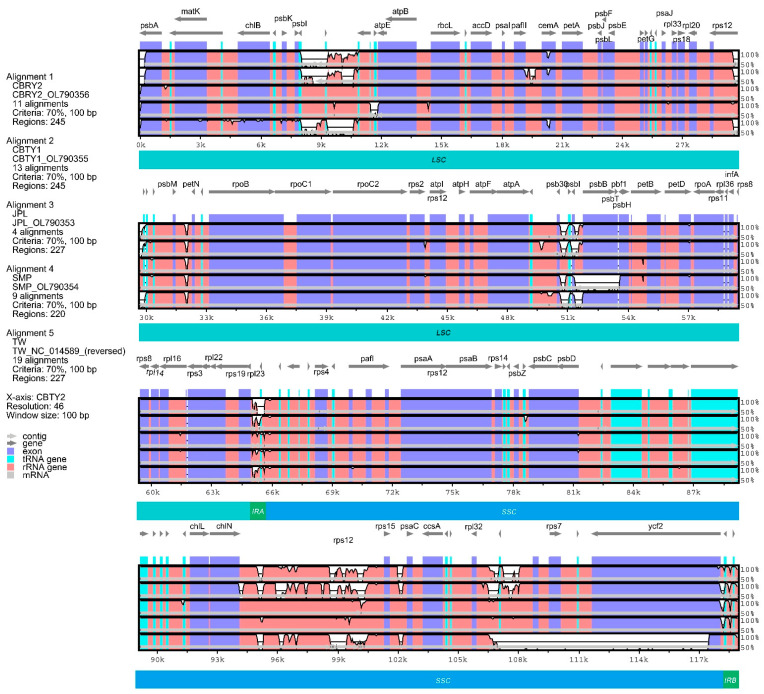
Comparative analysis of chloroplast genome sequence differences between *Cathaya argyrophylla* populations located at six different locations in China. The CBTY2 genome was used as the reference genome. Gray arrows and thick black lines above each alignment indicate gene orientation. The *y*-axis represents the percentage identity (50–100%). CBTY1, CBTY2 and CBRY2 were collected from Chengbu County of Hunan Province. JPL and SMP were collected from Chengzhou City of Hunan Province. TW was collected from Taipei City of Taiwan Province in a previously published study.

**Figure 5 genes-13-01963-f005:**
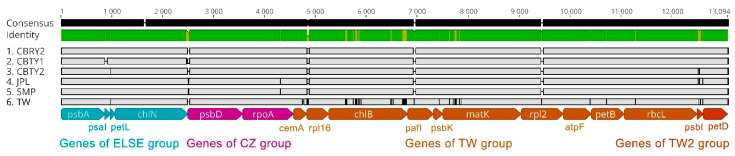
The alignment of polymorphic chloroplast genes between *Cathaya argyrophylla* populations located at different locations in China. Each gene is marked by different colored bars below the alignment. Each color represents a specific group classification for one unique polymorphic type. CBTY1, CBTY2 and CBRY2 were collected from Chengbu County of the Hunan Province. JPL and SMP were collected from Chengzhou City of the Hunan Province. TW was obtained from a previously published study, in which the samples were collected from Taipei City of the Taiwan Province.

**Figure 6 genes-13-01963-f006:**
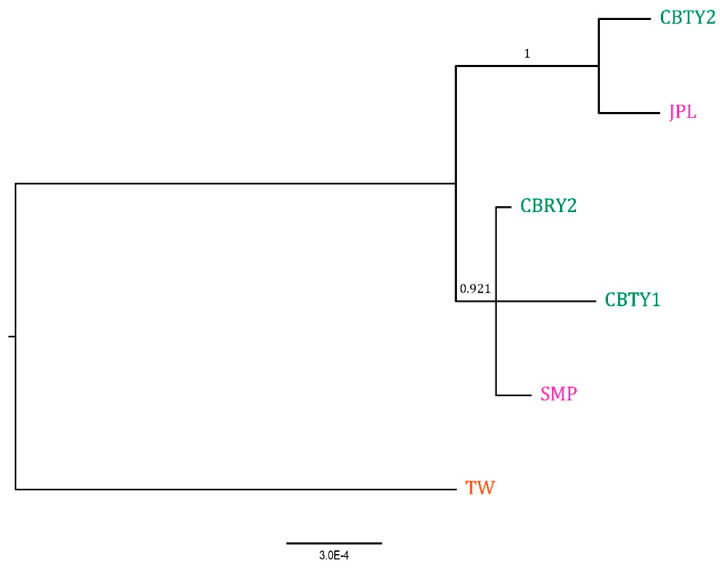
Phylogenetic analysis of C. argyrophylla populations in different locations of China. Bayes posterior probabilities are shown next to the nodes. The fonts in purple colors indicate populations from Chenzhou city; those in green colors indicate populations from Chengbu county; the only font in orange color indicates the population from Taiwan province. The number at the bottom of the figure is used for measuring genetic distance. CBTY1, CBTY2 and CBRY2 were collected from Chengbu County of the Hunan Province. JPL and SMP were collected from Chengzhou City of the Hunan Province. TW was obtained from a previously published study, in which the samples were collected from Taipei City of the Taiwan Province.

**Figure 7 genes-13-01963-f007:**
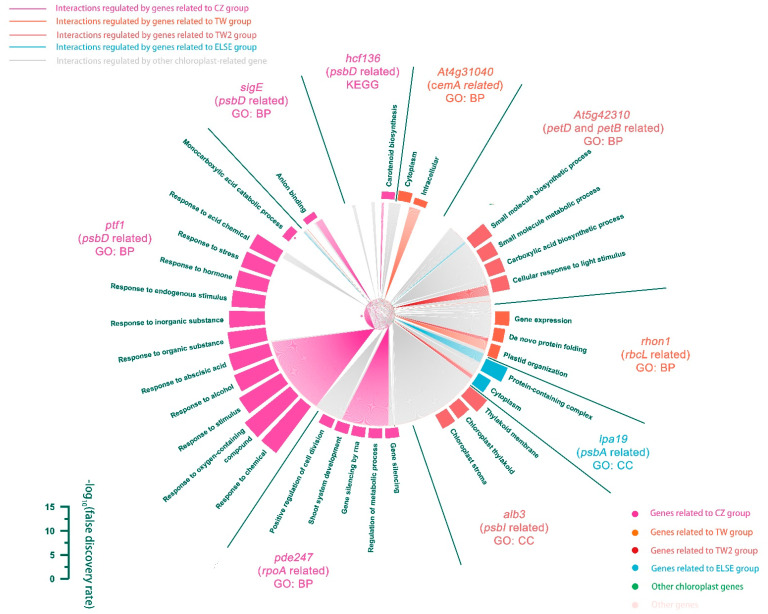
Gene regulatory network for the regulation between *Arabidopsis thaliana* chloroplast-related genes and its other genome-wide genes. The inner portion is the gene regulatory network, with different nodes representing genes related to different groups and different colored lines representing interactions regulated by genes related to the different groups. The outer portion is the circular barplot describing the functional enrichment (significantly enriched GO or KEGG terms) results of the below genes connected with the same colored lines as the bars. The regulators related to the polymorphic chloroplast genes of *C. argyrophylla* and the functional enrichment type of their regulated genes are marked by labels with different colors corresponding to their group classifications. CC indicates cellular component; BP indicates biological process and MF indicates molecular function.

**Figure 8 genes-13-01963-f008:**
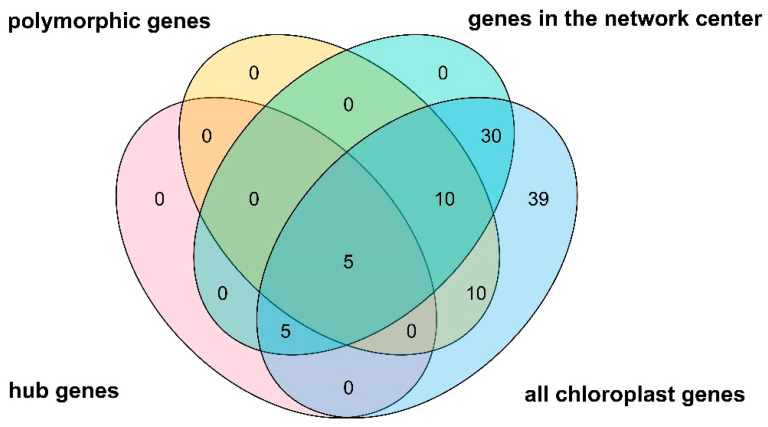
Venn diagram of all chloroplast genes included in the gene regulatory network for *Arabidopsis thaliana* chloroplast-related genes and its other genome-wide genes.

**Figure 9 genes-13-01963-f009:**
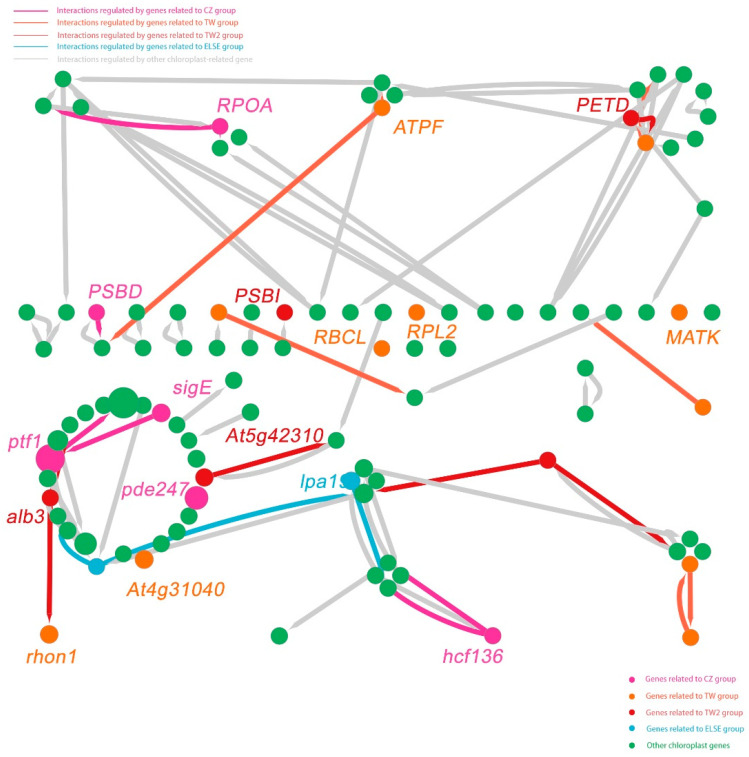
Gene regulatory network for *A. thaliana* chloroplast-related gene regulation. Nodes represent the genes related to different groups. Line color represents interactions regulated by genes in the different groups. Genes in uppercase letters represent location-related, polymorphic, homologous *C. argyrophylla* chloroplast genes, and genes in lowercase letters represent the literature-reported related chloroplast genes of those in uppercase letters.

**Table 1 genes-13-01963-t001:** Chloroplast genome features of *Cathaya argyrophylla* populations from six locations in China.

Category		CBTY1	CBTY2	CBRY2	JPL	SMP	TW
total genome	length (bp)	118,538	118,886	120,328	118,724	119,080	107,122
	GC content (%)	39	39	39	39	39	39
LSC	length (bp)	65,454	648,39	66,969	64,935	65,069	64,283
	GC content (%)	38	38	38	38	38	38
SSC	length (bp)	52,203	53,193	52,488	52,947	53,131	42,143
	GC content (%)	39	39	39	39	39	40
IR	length (bp)	882	854	872	842	880	782
	GC content (%)	34	39	35	38	39	36
number of total genes		112	108	112	109	109	110
number of CDS		73	70	73	71	70	70
number of tRNA genes		35	34	35	34	35	32
number of rRNA genes		4	4	4	4	4	4
Accession number		OL790355	OL753660	OL790356	OL790353	OL790354	NC014589

Note: CBTY1, CBTY2 and CBRY2 were collected from Chengbu County of Hunan Province. JPL and SMP were collected from Chengzhou City of Hunan Province. TW was collected from Taipei City of Taiwan Province in a previously published study.

**Table 2 genes-13-01963-t002:** Information regarding the polymorphic chloroplast genes between *Cathaya argyrophylla* populations located at six different locations in China.

Gene Name	Description	Location	Polymorphism Type	Group
*psbA*	Photosystem II protein D1 2	All locations	CBRY2, JPL, SMP, TW, CBTY2|CBTY1	ELSE
*psaI*	Photosystem I reaction centre subunit VIII	All locations	TW, CBTY2|JPL, CBRY2, CBTY1, SMP	ELSE
*petL*	Cytochrome b6/f complex subunit 6	All locations	TW, CBTY2|JPL, CBRY2, CBTY1, SMP	ELSE
*chlN*	Light-independent protochlorophyllide reductase subunit	All locations	CBTY1|TW, CBTY2, JPL, CBRY2, SMP	ELSE
*psbD*	Photosystem II D2 protein	All locations	JPL, SMP|TW, CBTY2, CBRY2, CBTY1	CZ
*rpoA*	DNA-directed RNA polymerase subunit α	All locations	JPL, SMP|TW, CBTY2, CBRY2, CBTY1	CZ
*ycf2*	Probable ATPase of unknown function	Only in hinterland	CBRY2|JPL, SMP|CBTY2, CBTY1	CZ
*cemA*	Chloroplast envelope membrane protein	All locations	TW|JPL, CBTY2, CBRY2, CBTY1, SMP	TW
*rpl16*	60S ribosomal protein L16-A	All locations	TW|JPL, CBTY2, CBRY2, CBTY1, SMP	TW
*chlB*	Light-independent protochlorophyllide reductase subunit B	All locations	TW|JPL, CBTY2, CBRY2, CBTY1, SMP	TW
*pafI*	Photosystem I assembly protein Ycf3	All locations	TW|JPL, CBTY2, CBRY2, CBTY1, SMP	TW
*psbK*	Photosystem II reaction centre protein K	All locations	TW|JPL, CBTY2, CBRY2, CBTY1, SMP	TW
*matK*	Chloroplast Maturase K	All locations	TW|JPL, CBTY2, CBRY2, CBTY1, SMP	TW
*rpl2*	50S ribosomal protein L2, chloroplastic	All locations	TW|JPL, CBTY2, CBRY2, CBTY1, SMP	TW
*atpF*	ATP synthase subunit b, chloroplastic	All locations	TW|JPL, CBTY2, CBRY2, CBTY1, SMP	TW
*petB*	Component of the cytochrome b6-f complex	All locations	TW|JPL, CBTY2, CBRY2, CBTY1, SMP	TW
*rbcL*	Ribulose bisphosphate carboxylase large chain	All locations	TW|JPL, CBTY2, CBRY2, CBTY1, SMP	TW
*psbI*	Photosystem II reaction centre protein I	All locations	TW|JPL, CBTY2|CBRY2, CBTY1, SMP	TW2
*petD*	Cytochrome b6-f complex subunit 4	All locations	TW|JPL, CBTY2|CBRY2, CBTY1, SMP	TW2

Note: CBTY1, CBTY2 and CBRY2 were collected from Chengbu County of the Hunan Province. JPL and SMP were collected from Chengzhou City of the Hunan Province. TW was obtained from a previously published study, in which the samples were collected from Taipei City of the Taiwan Province. Samples separated by the |symbol have a common polymorphism type for one gene.

## Data Availability

The complete chloroplast genome sequences of *C. argyrophylla* have been deposited in the GenBank database of National Center for Biotechnology Information (https://www.ncbi.nlm.nih.gov/) (accessed on 11 October 2021), and the accession numbers for CBTY1, CBTY2, CBRY2, SMP and JPL are OL790355, OL753660, OL790356, OL790354 and OL790353 respectively. The associated Bio Project for all samples is (PRJNA867043); the Bio-Sample numbers are SAMN30183339, SAMN30183340, SAMN30183343, SAMN30183342 and SAMN30183341, respectively; the SRA numbers are SRR20981869, SRR20981868, SRR20981867, SRR20981865 and SRR20981866, respectively.

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
