# Peer review of "Differences in the Chloroplast Genome and Its Regulatory Network among Cathaya argyrophylla Populations from Different Locations in China"

_genes, 2022, doi:10.3390/genes13111963_

Round 1

Reviewer 1 Report

I have gone through the manuscript in details. 

The author has to explain in details the relevance or the importance of the study on the introduction.

The Abstract has to be modified with more emphasis on the result and the interpretation

Spell check and grammar check has to be done by the authors

What does this sentence refer to?

All genomes were completed using the mVISTA soft- 102 ware [18] with the genome from CBTY2 as a reference.

The scientific name should be represented uniformly throughout the manuscript. The authors have not maintained uniformity

Regarding the figures the labels should start with a capital letter in all the images. The uniformity was maintained?

The authors have to experimentally validate to confirm the expression of a few genes that are unique to different locations and the results have to be incorporated in the manuscript for an additional validation. Experimental validation is missing in the manuscript.

The authors have to perform a detailed phylogenetic analysis to identify the phylogenetic relationship of the species in different location.

Reviewer 2 Report

I have thoroughly reviewed the paper entitled “Differences in the chloroplast genome and its regulatory network among
Cathaya argyrophylla populations from different locations in China
Journal”. The following minor corrections are needed to be incorporated to further improve the manuscript:

1.     The paper is written well and well formatted.

2.     Citations must be formatted uniformly like et al must be formatted accordingly.

3.     Scientific names must be italicized throughout the paper.

5.     The styling and format of references also need some attention.

Round 2

Reviewer 1 Report

Recommended to accept the manuscript